# On the Virasoro fusion and modular kernels at any irrational central charge

**Julien Roussillon**

*Department of Mathematics and Systems Analysis*
*P.O. Box 11100, FI-00076, Aalto University, Finland.*

*E-mail:* julien.roussillon@aalto.fi

ABSTRACT: We propose a series representation for the Virasoro fusion and modular kernels at any irrational central charge. Two distinct, yet closely related formulas are needed for the cases $c \in \mathbb{C}\backslash(-\infty, 1]$ and $c < 1$. We also conjecture that the formulas have a well-defined limit as the central charge approaches rational values. Our proposal for $c < 1$ agrees numerically with the fusion transformation of the four-point spherical conformal blocks, whereas our proposal for $c \in \mathbb{C}\backslash(-\infty, 1]$ agrees numerically with Ponsot and Teschner's integral formula for the fusion kernel. The case of the modular kernel is studied as a special case of the fusion kernel.

# 1 Introduction and main results

## 1.1 Introduction

Virasoro conformal blocks play a fundamental role in the conformal bootstrap approach to Conformal Field Theory in two dimensions. They are parts of correlation functions which are entirely determined by conformal symmetry. Consequently, they are special functions determined by representation theory of the Virasoro algebra.

On a given Riemann surface, Virasoro conformal blocks form certain bases for the solution space of the Virasoro Ward identities [19]. There exist linear transformations relating different bases of this space called *crossing transformations*. In general, they are integral transformations whose kernels are called *Virasoro crossing kernels*. A

prototypical example is a special case where the crossing transformations on the four-point Riemann sphere reduce to the well-known connection formulas for the Gauss hypergeometric function [9].

Of special importance is the case of the four-point Riemann sphere and of the one-point torus, because these two cases generate the set of all crossing kernels. The corresponding crossing kernels are denoted Virasoro fusion kernel and Virasoro modular kernels, respectively. Understanding the crossing properties of conformal blocks on these surfaces is primordial in the conformal bootstrap approach, since for instance the crossing symmetry equations for the correlation functions can be written in terms of the crossing kernels only [19]. For a recent review of Virasoro conformal blocks, their crossing properties and connections to other areas of physics and mathematics, the reader is referred to [4, 19].

Let us now describe more precisely the different objects at play. We will use the following parametrization of the conformal dimensions and central charge:

$$\Delta(P) = \frac{Q^2}{4} + P^2, \qquad c = 1 + 6Q^2, \qquad Q = b + \frac{1}{b}.$$

In the case $c \in \mathbb{C}\backslash(-\infty, 1]$, the Virasoro fusion kernel $\mathbf{F}$ is defined by the following relation:

$$\mathcal{F}_{P_s}^{(b)} \begin{bmatrix} P_2 & P_3 \\ P_1 & P_4 \end{bmatrix} (z) = \int_{\mathbb{R}} dP_t \ \mathbf{F}_{P_s, P_t}^{(b)} \begin{bmatrix} P_2 & P_3 \\ P_1 & P_4 \end{bmatrix} \mathcal{F}_{P_t}^{(b)} \begin{bmatrix} P_2 & P_3 \\ P_1 & P_4 \end{bmatrix} (1 - z), \qquad (1.1)$$

where $z$ is the cross-ratio of four points on the Riemann sphere, and where the conformal blocks $\mathcal{F}$ are defined in the natural normalization

$$\mathcal{F}_{P_s}^{(b)} \begin{bmatrix} P_2 & P_3 \\ P_1 & P_4 \end{bmatrix} (z) = z^{\Delta(P_s) - \Delta(P_1) - \Delta(P_2)} \left(1 + O(z)\right), \quad \text{as } z \to 0. \qquad (1.2)$$

The AGT correspondence [1] provides an explicit power series representation in $z$ for $\mathcal{F}$. There also exist recursion relations for $\mathcal{F}$ due to Zamolodchikov [30, 31] which converge much faster than the AGT formula (see also [19, section 2.4.2]).

Throughout the paper we introduce the convention $s(a \pm b) = s(a + b)s(a - b)$. $\mathbf{F}$ admits the following integral formula due to Ponsot and Teschner [17, 18]:

$$\mathbf{F}_{P_s, P_t}^{(b)} \begin{bmatrix} P_2 & P_3 \\ P_1 & P_4 \end{bmatrix} = \frac{\Gamma_b(Q \pm 2iP_s)}{\Gamma_b(\pm 2iP_t)} \frac{\Gamma_b(\frac{Q}{2} - iP_2 \pm iP_3 \pm iP_t)\Gamma_b(\frac{Q}{2} + iP_4 \pm iP_1 \pm iP_t)}{\Gamma_b(\frac{Q}{2} - iP_2 \pm iP_1 \pm iP_s)\Gamma_b(\frac{Q}{2} + iP_4 \pm iP_3 \pm iP_s)}$$

$$\times \int_{i\mathbb{R}} du \ \frac{S_b(\frac{Q}{4} - iP_2 \pm iP_1 + u)S_b(\frac{Q}{4} + iP_4 \pm iP_3 + u)}{S_b(\frac{3Q}{4} - iP_2 + iP_4 \pm iP_t + u)S_b(\frac{3Q}{4} \pm iP_s + u)}. \qquad (1.3)$$

Here we use the standard notation for Barnes' double Gamma function $\Gamma_b(x)$, and the double Sine function $S_b(x) = \frac{\Gamma_b(x)}{\Gamma_b(Q-x)}$, see [4, Appendix B] for a review of their properties.

It is well known that the Ponsot-Teschner formula (1.3) satisfies a set of shift equations in its momenta which originates from the pentagon relation [4, 24]. Eberhardt showed in [4] that $\mathbf{F}$ is the unique solution of the shift equations for $b^2 \notin \mathbb{Q}$ and $c \in \mathbb{C}\backslash(-\infty, 1]$ that is meromorphic in all of its parameters. However, $\mathbf{F}$ does not admit an analytic continuation to $c \leq 1$, since the function $\Gamma_b$ is not defined for $b \in i\mathbb{R}$. Interestingly, Ribault and Tsiares discovered in [23] a transformation, the Virasoro-Wick rotation, which maps the unique meromorphic solution of the shift equations in the regime $c \in \mathbb{C}\backslash(-\infty, 1]$ to the unique meromorphic solution for $c \in \mathbb{C}\backslash[25, \infty)$. However, the image of $\mathbf{F}$ under this transformation is an odd function of $P_s$ and $P_t$, hence it cannot be the physical fusion kernel for $c < 1$ since the conformal blocks are even. In view of Eberhardt's uniqueness result, Ribault and Tsiares were led to the conclusion that $\hat{\mathbf{F}}$ must have weaker analyticity properties than $\mathbf{F}$.

In what follows, we denote by $\hat{\mathbf{F}}$ the Virasoro fusion kernel for $c \leq 1$, or, equivalently, for $b \in i\mathbb{R}$. It will be convenient to describe it in terms of another set of parameters

$$\beta = ib, \qquad p = iP, \qquad \hat{Q} = \beta + \frac{1}{\beta}. \tag{1.4}$$

Then, we define $\hat{\mathbf{F}}$ to be such that

$$\mathcal{F}^{(b)}_{P_s}\begin{bmatrix} P_2 & P_3 \\ P_1 & P_4 \end{bmatrix}(z) = \int_{i\mathbb{R}+\Lambda} dp_t\, \hat{\mathbf{F}}^{(\beta)}_{p_s,p_t}\begin{bmatrix} p_2 & p_3 \\ p_1 & p_4 \end{bmatrix} \mathcal{F}^{(b)}_{P_t}\begin{bmatrix} P_2 & P_3 \\ P_1 & P_4 \end{bmatrix}(1-z). \tag{1.5}$$

The choice of parameter dependence for $\hat{\mathbf{F}}$ will be justified in section 2. Note that in this case we are forced to shift the contour of integration by $\Lambda \in \mathbb{R}^*$ [22, 23] because the conformal blocks on the right-hand side have poles at

$$p_t^{(m,n)} = \frac{i}{2}(m\beta + n\beta^{-1}), \qquad m,n \in \mathbb{N}^*, \tag{1.6}$$

hence the poles lie on the imaginary axis for $\beta \in \mathbb{R}$. Moreover, the result of the integral should not depend on $\Lambda$.

Finally, we define the Virasoro modular kernels $\mathbf{M}$ and $\hat{\mathbf{M}}$ to be the following special cases of $\mathbf{F}$ and $\hat{\mathbf{F}}$ [10]:

$$\mathbf{M}^{(b)}_{P_s,P_t}[P_0] = \sqrt{2}\, 256^{P_t^2 - P_s^2}\, \mathbf{F}^{(\sqrt{2}b)}_{\sqrt{2}P_s,\sqrt{2}P_t}\begin{bmatrix} \frac{P_0}{\sqrt{2}} & \frac{ib}{2\sqrt{2}} \\ \frac{ib}{2\sqrt{2}} & \frac{ib}{2\sqrt{2}} \end{bmatrix}, \tag{1.7}$$

$$\hat{\mathbf{M}}^{(\beta)}_{p_s,p_t}[p_0] = \sqrt{2}\, 256^{-p_t^2 + p_s^2}\, \hat{\mathbf{F}}^{(\sqrt{2}\beta)}_{\sqrt{2}p_s,\sqrt{2}p_t}\begin{bmatrix} \frac{p_0}{\sqrt{2}} & \frac{i\beta}{2\sqrt{2}} \\ \frac{i\beta}{2\sqrt{2}} & \frac{i\beta}{2\sqrt{2}} \end{bmatrix}. \tag{1.8}$$

This originates from Poghossian's observation that the one-point toric conformal blocks are special cases of the four-point spherical conformal blocks [16].

## 1.2 Main results

In this paper, we propose a series representation for both $\mathbf{F}$ and $\hat{\mathbf{F}}$ as well as for the modular kernels $\mathbf{M}$ and $\hat{\mathbf{M}}$. As a matter of convenience for the reader, all formulas are gathered below.

### 1.2.1 The fusion kernels

Our proposals for $\mathbf{F}$ and $\hat{\mathbf{F}}$ are as follows:

$$\mathbf{F}^{(b)}_{P_s,P_t} \begin{bmatrix} P_2 & P_3 \\ P_1 & P_4 \end{bmatrix} = \frac{1}{2}\left(\mathbf{F}^{+,(b)}_{P_s,P_t}\begin{bmatrix} P_2 & P_3 \\ P_1 & P_4 \end{bmatrix} + \mathbf{F}^{-,(b)}_{P_s,P_t}\begin{bmatrix} P_2 & P_3 \\ P_1 & P_4 \end{bmatrix}\right), \tag{1.9}$$

$$\hat{\mathbf{F}}^{(\beta)}_{p_s,p_t}\begin{bmatrix} p_2 & p_3 \\ p_1 & p_4 \end{bmatrix} = \frac{1}{2}\left(\hat{\mathbf{F}}^{+,(\beta)}_{p_s,p_t}\begin{bmatrix} p_2 & p_3 \\ p_1 & p_4 \end{bmatrix} + \hat{\mathbf{F}}^{-,(\beta)}_{p_s,p_t}\begin{bmatrix} p_2 & p_3 \\ p_1 & p_4 \end{bmatrix}\right), \tag{1.10}$$

where $\mathbf{F}^{-,(b)}_{P_s,P_t} = \mathbf{F}^{+,(b)}_{-P_s,P_t}$, $\hat{\mathbf{F}}^{-,(\beta)}_{p_s,p_t} = \hat{\mathbf{F}}^{+,(\beta)}_{-p_s,p_t}$, and where $\mathbf{F}^+$ and $\hat{\mathbf{F}}^+$ take the form

$$\mathbf{F}^{+,(b)}_{P_s,P_t}\begin{bmatrix} P_2 & P_3 \\ P_1 & P_4 \end{bmatrix} = K^{(b)}_{P_s,P_t}\begin{bmatrix} P_2 & P_3 \\ P_1 & P_4 \end{bmatrix} f^{(b)}_{P_s,P_t}\begin{bmatrix} P_2 & P_3 \\ P_1 & P_4 \end{bmatrix}, \tag{1.11}$$

$$\hat{\mathbf{F}}^{+,(\beta)}_{p_s,p_t}\begin{bmatrix} p_2 & p_3 \\ p_1 & p_4 \end{bmatrix} = \hat{K}^{(\beta)}_{p_s,p_t}\begin{bmatrix} p_2 & p_3 \\ p_1 & p_4 \end{bmatrix} f^{(\beta)}_{p_s,p_t}\begin{bmatrix} p_2 & p_3 \\ p_1 & p_4 \end{bmatrix}. \tag{1.12}$$

The factors $K$ and $\hat{K}$ are given by

$$K^{(b)}_{P_s,P_t}\begin{bmatrix} P_2 & P_3 \\ P_1 & P_4 \end{bmatrix} = e^{i\pi\left(P_1^2+P_2^2+P_3^2+P_4^2+\frac{1+b^2+b^{-2}}{4}\right)} \tag{1.13}$$

$$\times \frac{\Gamma_b(2iP_s)\Gamma_b(Q+2iP_s)}{\Gamma_b(-2iP_t)\Gamma_b(Q-2iP_t)} \frac{\Gamma_b(\frac{Q}{2}-iP_t\pm iP_2\pm iP_3)\Gamma_b(\frac{Q}{2}-iP_t\pm iP_1\pm iP_4)}{\Gamma_b(\frac{Q}{2}+iP_s\pm iP_2\pm iP_1)\Gamma_b(\frac{Q}{2}+iP_s\pm iP_3\pm iP_4)},$$

$$\hat{K}^{(\beta)}_{p_s,p_t}\begin{bmatrix} p_2 & p_3 \\ p_1 & p_4 \end{bmatrix} = -ie^{i\pi\left(p_1^2+p_2^2+p_3^2+p_4^2+\frac{1+\beta^2+\beta^{-2}}{4}\right)} \tag{1.14}$$

$$\times \frac{\Gamma_\beta(2ip_t+\frac{1}{\beta})\Gamma_\beta(2ip_t+\beta)}{\Gamma_\beta(-2ip_s+\frac{1}{\beta})\Gamma_\beta(-2ip_s+\beta)} \frac{\Gamma_\beta(\frac{\hat{Q}}{2}-ip_s\pm ip_2\pm ip_1)\Gamma_\beta(\frac{\hat{Q}}{2}-ip_s\pm ip_3\pm ip_4)}{\Gamma_\beta(\frac{\hat{Q}}{2}+ip_t\pm ip_2\pm ip_3)\Gamma_\beta(\frac{\hat{Q}}{2}+ip_t\pm ip_1\pm ip_4)}.$$

It remains to describe $f$ which is an infinite series of the form

$$f^{(b)}_{P_s,P_t}\begin{bmatrix} P_2 & P_3 \\ P_1 & P_4 \end{bmatrix} := e^{2i\pi P_s P_t}\sum_{k=0}^{\infty}\sum_{l=0}^{\infty}\alpha_k(b,P_s)\alpha_l(\tfrac{1}{b},P_s)e^{-2\pi bkP_t}e^{-\frac{2\pi lP_t}{b}}. \tag{1.15}$$

We provide a recursive representation for the coefficients $\alpha_n$:

$$\alpha_n(b,P_s) = \delta_{n,0} \tag{1.16}$$

$$+ \sum_{l=1}^{n} e^{-\pi b(P_s+\frac{lib}{2})}\left[\frac{\phi_{ln}(P_1,P_2,P_3,P_4)}{\sinh(\pi b(P_s+\frac{lib}{2}))} - \frac{\phi_{ln}(P_1,P_2+\frac{i}{2b},P_3+\frac{i}{2b},P_4)}{\cosh(\pi b(P_s+\frac{lib}{2}))}\right].$$

The coefficients $\phi_{nn}$, $n > 0$ have the explicit form

$$\phi_{nn}(P_1,P_2,P_3,P_4) = \frac{i2^{2n-1}(-1)^{n+1}}{\sin(n\pi b^2)\prod_{l=1}^{n-1}\sin(l\pi b^2)^2} \tag{1.17}$$

$$\times \prod_{l=1}^{n}\cosh\left(\pi b\left(\pm P_1+P_2+ib\left(l-\tfrac{n+1}{2}\right)\right)\right)\cosh\left(\pi b\left(P_3\pm P_4+ib(l-\tfrac{n+1}{2})\right)\right),$$

whereas the coefficients $\phi_{ln}$ for $0 < l \leq n$ have the semi-explicit form

$$\phi_{ln}(P_1,P_2,P_3,P_4) = \phi_{ll}(P_1,P_2,P_3,P_4)\,\alpha_{n-l}\left(b,\frac{lib}{2}\right). \tag{1.18}$$

For instance, $\alpha_1$ explicitly reads

$$\alpha_1(b, P_s) = \frac{2ie^{-\pi b(Ps + \frac{ib}{2})}}{\sin(\pi b^2)} \tag{1.19}$$

$$\times \left( \frac{\cosh(\pi b(P_2 \pm P_1))\cosh(\pi b(P_3 \pm P_4))}{\sinh(\pi b(P_s + \frac{ib}{2}))} - \frac{\sinh(\pi b(P_2 \pm P_1))\sinh(\pi b(P_3 \pm P_4))}{\cosh(\pi b(P_s + \frac{ib}{2}))} \right).$$

More generally, utilizing (1.16) inductively, $\phi_{ln}$ can be expressed only in terms of $\phi_{mm}(P_1, P_2, P_3, P_4)$ and $\phi_{mm}(P_1, P_2 + \frac{i}{2b}, P_3 + \frac{i}{2b}, P_4)$ for $m = 1, ..., n - l$.

### 1.2.2 The modular kernels

We now proceed with the case of the modular kernels. We verified numerically that the formulas below for $\mathbf{M}$ and $\hat{\mathbf{M}}$ satisfy (1.7) and (1.8), respectively. We have

$$\mathbf{M}^{(b)}_{P_s,P_t}[P_0] = \frac{1}{2}\left(\mathbf{M}^{+,(b)}_{P_s,P_t}[P_0] + \mathbf{M}^{-,(b)}_{P_s,P_t}[P_0]\right), \tag{1.20}$$

$$\hat{\mathbf{M}}^{(\beta)}_{p_s,p_t}[p_0] = \frac{1}{2}\left(\hat{\mathbf{M}}^{+,(\beta)}_{p_s,p_t}[p_0] + \hat{\mathbf{M}}^{-,(\beta)}_{p_s,p_t}[p_0]\right), \tag{1.21}$$

where $\mathbf{M}^{-,(b)}_{P_s,P_t} = \mathbf{M}^{+,(b)}_{-P_s,P_t}$, $\hat{\mathbf{M}}^{-,(\beta)}_{p_s,p_t} = \hat{\mathbf{M}}^{+,(\beta)}_{-p_s,p_t}$, and

$$\mathbf{M}^{+,(b)}_{P_s,P_t}[p_0] = L^{(b)}_{P_s,P_t}[P_0] \, g^{(b)}_{P_s,P_t}[P_0], \tag{1.22}$$

$$\hat{\mathbf{M}}^{+,(\beta)}_{p_s,p_t}[p_0] = \hat{L}^{(\beta)}_{p_s,p_t}[p_0] \, g^{(\beta)}_{p_s,p_t}[p_0]. \tag{1.23}$$

The factors $L$ and $\hat{L}$ read

$$L^{(b)}_{P_s,P_t}[P_0] = \sqrt{2} \, e^{\frac{i\pi}{2}(\frac{Q^2}{4}+P_0^2)} \frac{\Gamma_b(2iP_s)\Gamma_b(Q+2iP_s)}{\Gamma_b(-2iP_t)\Gamma_b(Q-2iP_t)} \frac{\Gamma_b(\frac{Q}{2} \pm iP_0 - 2iP_t)}{\Gamma_b(\frac{Q}{2} \pm iP_0 + 2iP_s)}, \tag{1.24}$$

$$\hat{L}^{(\beta)}_{p_s,p_t}[p_0] = -i\sqrt{2}e^{\frac{i\pi}{2}(\frac{\hat{Q}^2}{4}+p_0^2)} \frac{\Gamma_\beta(2ip_t + \frac{1}{\beta})\Gamma_\beta(2ip_t + \beta)}{\Gamma_\beta(-2ip_s + \frac{1}{\beta})\Gamma_\beta(-2ip_s + \beta)} \frac{\Gamma_\beta(\frac{\hat{Q}}{2} \pm ip_0 - 2ip_s)}{\Gamma_\beta(\frac{\hat{Q}}{2} \pm ip_0 + 2ip_t)}. \tag{1.25}$$

Finally, the series is of the form

$$g^{(b)}_{P_s,P_t}[P_0] = e^{4i\pi P_s P_t} \sum_{k=0}^{\infty} \sum_{l=0}^{\infty} \mu_k(b, P_s)\mu_l(\tfrac{1}{b}, P_s)e^{-4\pi bkP_t}e^{-\frac{4\pi lP_t}{b}}, \tag{1.26}$$

and the coefficients $\mu_k$ have an explicit form in terms of $q$-Pochhammer symbols. More precisely, let

$$(a, q)_n = \prod_{k=0}^{n-1}(1 - aq^k). \tag{1.27}$$

and denote $\alpha_0 := \frac{Q}{2} + iP_0$. Then, we have

$$\mu_n(b, P_s) \tag{1.28}$$

$$= \sum_{l=0}^{n} \frac{\left(e^{2i\pi b\alpha_0}; e^{2i\pi b^2}\right)_l}{\left(e^{2i\pi b^2}; e^{2i\pi b^2}\right)_l} \frac{\left(e^{-2\pi b(2P_s + i\alpha_0)}; e^{-2i\pi b^2}\right)_l}{\left(e^{-2\pi b(2P_s + ib)}; e^{-2i\pi b^2}\right)_l} \frac{\left(e^{-2i\pi b\alpha_0}; e^{2i\pi b^2}\right)_{n-l}}{\left(e^{2i\pi b^2}; e^{2i\pi b^2}\right)_{n-l}} e^{2i\pi b\alpha_0(n-l)}.$$

## 1.3 Numerical tests

The details of the numerical tests described in this section can be found in the ancillary Jupyter notebook, or in the GitLab repository [25].

### 1.3.1 The fusion transformation for $c < 1$

We performed a test of the fusion transformation (1.5) with the following values of the parameters:

$$\begin{pmatrix} \beta \\ p_1 \\ p_2 \\ p_3 \\ p_4 \end{pmatrix} = \begin{pmatrix} 0.6 \\ 0.4i \\ 0.6i \\ 0.3i \\ 0.5i \end{pmatrix}, \qquad p_s = 0.35i, \quad z = 0.5, \quad \Lambda = 0.5. \tag{1.29}$$

We used the publicly available GitLab repository [21] developed by Ribault to compute values of the function $\Gamma_b$ and of the conformal blocks.

The accuracy of the computations is controlled by three parameters $(T, L, N_{\max})$.[1] The parameter $T$ represents the truncation of the two series in f (1.15) at order $T$. $L$ represents the truncation of the infinite integration line $i\mathbb{R} + \Lambda$ to $i[-L, L] + \Lambda$. Finally, $N_{\max}$ corresponds to the truncation of Zamolodchikov's recursion relation for the conformal blocks [21]. At the values (1.29), the left-hand side of (1.5) is real-valued and equals approximatively 1.08. Then, we made the following verifications:

| $(T, N_{\max}, L)$ | Accuracy of Re (l.h.s - r.h.s of (1.5)) |
|---|---|
| (0,10,1) | $9.0 \times 10^{-1}$ |
| (1,12,2) | $1.3 \times 10^{-1}$ |
| (2,14,3) | $4.0 \times 10^{-4}$ |
| (3,16,4) | $1.3 \times 10^{-5}$ |
| (4,18,5) | $9.0 \times 10^{-10}$ |
| (6,20,6) | $8.3 \times 10^{-16}$ |
| (8,25,8) | $2.1 \times 10^{-16}$ |

.

We do not write Im (l.h.s - r.h.s of (1.5)) because it has a similar order of magnitude. Notice that we chose $\Lambda = 0.5$ to stay sufficiently far away from the poles of the integrand which lie on the imaginary axis. This has the effect of reducing the amplitude of the peaks and the oscillations of the integrand in (1.5), thereby increasing accuracy.

We were also able to verify the fusion transformation[2] for $\Lambda = 0.05$ and with parameters $(T, N_{\max}, L) = (8, 20, 2)$ with an accuracy of $9 \times 10^{-7}$. It appears challenging

---

[1]To reach an accuracy of $\sim 10^{-16}$ we also decreased slighlty the value of the parameters epsabs and epsrel of the Python method quad. We do not mention it in the main text for convenience.

[2]In this case we could not use the quad method due to high oscillations of the integrand. We used the method CubicSpline.integrate instead.

to increase accuracy in this case, because the integrand has rather high oscillations. As an illustration, the real parts of the integrand (1.5) for $\Lambda = 0.5$ and $\Lambda = 0.05$ are shown in Figure 1.

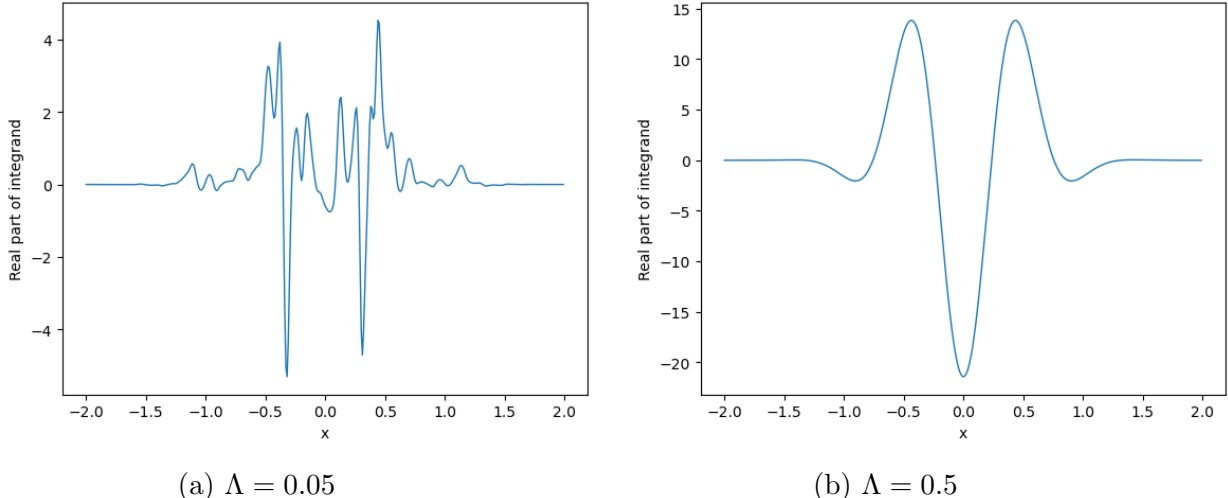

(a) $\Lambda = 0.05$            (b) $\Lambda = 0.5$

**Figure 1**: Plot of the real part of the integrand in (1.5) with $p_t = ix + \Lambda$.

Finally, we also verified numerically that the components $\hat{\mathbf{F}}^+$ and $\hat{\mathbf{F}}^-$ individually verify the fusion transformation (1.5)[3]. This implies that the integration of the t-channel conformal blocks against $\hat{\mathbf{F}}^+ - \hat{\mathbf{F}}^-$ vanishes. This is an important consistency check, since the s-channel conformal blocks are even in $P_s$.

### 1.3.2 Comparing F with Ponsot and Teschner's formula

We also compared the proposal (1.9) and Ponsot-Teschner's formula (1.3) for the following values of the parameters:

$$\begin{pmatrix} b \\ P_1 \\ P_2 \\ P_3 \\ P_4 \end{pmatrix} = \begin{pmatrix} 0.77 \\ 0.5i \\ 0.6i \\ 0.18i \\ 0.31i \end{pmatrix}, \qquad P_s = 0.35, \quad P_t = 0.65.$$

In this case, the only parameter which controls the accuracy is the truncation order $T$ of the series. At these values of the parameters, the Ponsot-Teschner formula gives approximatively $0.308 + 5.22 \times 10^{-22}$. We then performed the following calculations:

---

[3]We thank Ioannis Tsiares for suggesting us to perform this check.

| T | Accuracy of Re $(2\times(1.9) - (1.3))$ |
|---|---|
| 0 | $3.3 \times 10^{-2}$ |
| 1 | $8.1 \times 10^{-4}$ |
| 3 | $5.8 \times 10^{-6}$ |
| 5 | $1.0 \times 10^{-8}$ |
| 7 | $1.8 \times 10^{-11}$ |
| 9 | $5.2 \times 10^{-14}$ |
| 12 | $9.0 \times 10^{-16}$ |

.

We also have similar orders of magnitude for the accuracy of the imaginary part. We finally notice that we needed to multiply (1.9) by 2 because the equality is verified in the domain $\text{Re}(P_t) > 0$, where the series is seen to converge. By evenness of the integrand in the fusion transformation (1.1), we then gain a factor of 2.

## 1.4 Relations to earlier works

In a nutshell, our strategy which led to the proposals was to find the correct ansatz for the series and to solve the shift equations. We emphasize that, by construction, the proposals for $\mathbf{F}^{(b)}_{P_s,P_t} \begin{bmatrix} P_2 & P_3 \\ P_1 & P_4 \end{bmatrix}$ and $\hat{\mathbf{F}}^{(ib)}_{iP_s,iP_t} \begin{bmatrix} iP_2 & iP_3 \\ iP_1 & iP_4 \end{bmatrix}$ satisfy the same shift equations, however, the two are not related by analytic continuation. A similar idea was pursued by Zamolodchikov in [32] and led him to the discovery of the structure constants of Liouville theory for $c \leq 1$.

In the physics literature, seeking series solutions of the shift equations was initiated by Nemkov in [13] in the case of $\mathbf{M}$, building on earlier works [6, 15] (see also [14] for ideas related to $\mathbf{F}$). It would be interesting to compare the formula for $\mathbf{M}$ with Nemkov's results. In the mathematics literature, Ruijsenaars derived a rigorous series representation for a renormalized version of $\mathbf{M}$ dubbed relativistic conical hypergeometric function (see [27, 29]), and our proposal for $\mathbf{M}$ is essentially a rewriting of his findings. In particular, Ruijsenaars already wrote the series (1.26) in [27, Equations (2.4), (2.59)]. He proved such a series representation by first considering the case where $P_0$ takes a set of values which is dense in the imaginary line, and then by taking the interpolation limit.

## 1.5 Discussion and outlook

Below, we mention several directions that deserve further investigation.

- It is important to understand better the analytic properties of the series f (1.15). Moreover, several key identities (we have in mind (2.13) and (3.13), as well as (2.12) for the coefficients of the series) are stated without proof, although they have been verified numerically up to high order. Let us mention that the one-point toric analogs of these three equations were proved by Ruijsenaars in [26, 27].

- Ruijsenaars rigorously proved in [28] the leading asymptotics of a renormalized version of the Ponsot-Teschner formula (1.3) as $\operatorname{Re} P_t \to +\infty$ (by evenness in $P_t$, the case $\operatorname{Re} P_t \to -\infty$ obviously follows). We verified that our formula (1.9) reduces to his findings when only the first term in the series $f$ is kept. More generally, we believe that (1.9) should be understood as a complete asymptotic expansion of (1.3) as $\operatorname{Re} P_t > \delta$ and for some sufficiently large $\delta$. By analogy with the case $c > 25$, it is tempting to believe that there exists a compact formula for the physical fusion kernel for $c < 1$ which is manifestly even in $P_t$ and whose asymptotic expansion corresponds to (1.10). If such a compact formula exists, it should have weaker analyticity properties than the Ponsot-Teschner formula.

- If $b^2 = \frac{M}{N}$ with $M$ and $N$ coprime integers, then it can be seen from (1.17) that the coefficients $\phi_{nn}$ (and, therefore, $\alpha_n(b, P_s)$) diverge for $n \geq N$. Similarly, in this case we also have that $\alpha_n(b^{-1}, P_s)$ diverges for $n \geq M$. However, rather surprisingly, it might well be that the series $f$ itself has a well-defined limit as $b^2 \to \frac{M}{N}$ [4]. For instance, in the case $b^2 = 1$ we observed that the first two terms in the series, namely,

$$\alpha_1(b, P_s)e^{-2\pi b P_t} + \alpha_1(b^{-1}, P_s)e^{-2\pi b^{-1} P_t},$$

$$\alpha_2(b, P_s)e^{-4\pi b P_t} + \alpha_2(b^{-1}, P_s)e^{-4\pi b^{-1} P_t} + \alpha_1(b, P_s)\alpha_1(b^{-1}, Ps)e^{-2\pi b P_t}e^{-2\pi b^{-1} P_t},$$

have a well-defined limit as $b \to 1$. Moreover, the numerical tests of the fusion transformation (1.5) were performed for $\beta^2 = \frac{9}{25}$. This indicate that the apparent divergence of the series $f$ at rational values of $\beta^2$ is merely an artifact of the formulas, and the limit should be well-defined.

  We then conjecture that the limit $\beta^2$ rational of $f$ is well-defined. If true, it would mean that the limits of $\mathbf{F}$ as $c \to 25$ and of $\hat{\mathbf{F}}$ as $c \to 1$ are well-defined. It would be interesting to compare such limits with the known fusion kernels. At $c = 1$, it is proportional to the connection constant for the Painlevé VI tau function [5, 11], whereas at $c = 25$ a non-integral representation was recently constructed by Ribault and Tsiares in [23].

- The limit (1.7) implies that the coefficients $\alpha_n$ in (1.16) reduce to $\mu_n$ (1.28). This is highly nontrivial and suggests that there exists a simpler, fully explicit representation for $\alpha_n$ in terms of $q$-Pochhammer symbols.

- In [7], Ghosal, Remy, Sun and Sun provided a rigorous probabilistic construction of the four-point spherical conformal blocks and proved the fusion transformation (1.1) for $c > 25$ in a certain region of the parameter space. The case of the one-point toric conformal blocks was also proved in [8]. It would be interesting to obtain our formulas for $\mathbf{F}$ and $\mathbf{M}$ within their framework.

---

[4] A similar observation was made by Ruijsenaars in the case of the series $g$ entering $\mathbf{M}$ [27].

## 2 Formal derivation of the result

In this section we explain how we arrived to the claims (1.9) and (1.10), and we discuss some of their properties.

### 2.1 Solving the renormalized shift equations

The Virasoro fusion kernel is known to satisfy two pairs of shift equations in $P_s$ and $P_t$ [24]. More precisely, define the shift operators

$$H_{P_s}^{(b)} \begin{bmatrix} P_2 & P_3 \\ P_1 & P_4 \end{bmatrix} := h(P_s)e^{ib\partial_{P_s}} + h(-P_s)e^{-ib\partial_{P_s}} + V_{P_s} \begin{bmatrix} P_2 & P_3 \\ P_1 & P_4 \end{bmatrix}, \tag{2.1}$$

$$\tilde{H}_{P_t}^{(b)} \begin{bmatrix} P_2 & P_3 \\ P_1 & P_4 \end{bmatrix} := \tilde{h}(P_t)e^{ib\partial_{P_t}} + \tilde{h}(-P_t)e^{-ib\partial_{P_t}} + V_{P_t} \begin{bmatrix} P_2 & P_1 \\ P_3 & P_4 \end{bmatrix}, \tag{2.2}$$

with $e^{\pm ib\partial_{P_s}} y(P_s) := y(P_s \pm ib)$ and where

$$h(P_s) = 4\pi^2 \frac{\Gamma(1 + 2b^2 - 2ibP_s)\Gamma(b^2 - 2ibP_s)\Gamma(-2ibP_s)\Gamma(1 + b^2 - 2ibP_s)}{\prod_{\epsilon,\epsilon'=\pm 1}\Gamma\left(\frac{bQ}{2} - ib(P_s + \epsilon P_3 + \epsilon' P_4)\right)\Gamma\left(\frac{bQ}{2} - ib(P_s + \epsilon P_1 + \epsilon' P_2)\right)},$$

$$\tilde{h}(P_t) = 4\pi^2 \frac{\Gamma(1 - b^2 + 2ibP_t)\Gamma(1 + 2ibP_t)\Gamma(2ibP_t - 2b^2)\Gamma(2ibP_t - b^2)}{\prod_{\epsilon,\epsilon'=\pm}\Gamma\left(\frac{1-b^2}{2} + ib(P_t + \epsilon P_1 + \epsilon' P_4)\right)\Gamma\left(\frac{1-b^2}{2} + ib(P_t + \epsilon P_3 + \epsilon' P_2)\right)},$$

and

$$V_{P_s} \begin{bmatrix} P_2 & P_3 \\ P_1 & P_4 \end{bmatrix} = -2\cosh\left(2\pi b(P_2 + P_3 + \tfrac{ib}{2})\right)$$

$$+ 4\sum_{k=\pm} \frac{\prod_{\epsilon=\pm}\cosh\left(\pi b(\epsilon P_4 - \frac{ib}{2} - P_3 - kP_s)\right)\cosh\left(\pi b(\epsilon P_1 - \frac{ib}{2} - P_2 - kP_s)\right)}{\sinh\left(2\pi b(kP_s + \frac{ib}{2})\right)\sinh\left(2\pi b kP_s\right)}. \tag{2.3}$$

Then, the Virasoro fusion kernel satisfies [24]

$$H_{P_s}^{(b^{\pm 1})} \mathbf{F}_{P_s,P_t}^{(b)} = 2\cosh(2\pi b^{\pm 1} P_t)\mathbf{F}_{P_s,P_t}^{(b)}, \tag{2.4}$$

$$\tilde{H}_{P_t}^{(b^{\pm 1})} \mathbf{F}_{P_s,P_t}^{(b)} = 2\cosh(2\pi b^{\pm 1} P_s)\mathbf{F}_{P_s,P_t}^{(b)}. \tag{2.5}$$

Ribault and Tsiares found in [23] that the image of $\mathbf{F}$ under the Virasoro-Wick rotation

$$\mathcal{R}\mathbf{F}_{P_s,P_t}^{(b)} \begin{bmatrix} P_2 & P_3 \\ P_1 & P_4 \end{bmatrix} := \frac{P_t}{P_s}\mathbf{F}_{iP_t,iP_s}^{(ib)} \begin{bmatrix} iP_2 & iP_1 \\ iP_3 & iP_4 \end{bmatrix} \tag{2.6}$$

satisfies the same shift equations. As described in the introduction, the issue is that $\mathcal{R}\mathbf{F}$ is odd in $P_s$ and $P_t$, hence it is a nonphysical solution of the shift equations.

Our strategy which leads to (1.9) and (1.10) is as follows. We introduce another pair of shift operators[5]

$$D_{P_s}^{(b)} \begin{bmatrix} P_2 & P_3 \\ P_1 & P_4 \end{bmatrix} := e^{-ib\partial_{P_s}} + A_{P_s}^{(b)} \begin{bmatrix} P_2 & P_3 \\ P_1 & P_4 \end{bmatrix} e^{ib\partial_{P_s}} + V_{P_s}^{(b)} \begin{bmatrix} P_2 & P_3 \\ P_1 & P_4 \end{bmatrix}, \tag{2.7}$$

$$\tilde{D}_{P_t}^{(b)} \begin{bmatrix} P_2 & P_3 \\ P_1 & P_4 \end{bmatrix} := D_{P_t}^{(b)} \begin{bmatrix} P_2 & P_1 \\ P_3 & P_4 \end{bmatrix}, \tag{2.8}$$

---

[5]The shift operators $\mathcal{D}$ and $\tilde{\mathcal{D}}$ were first introduced by Ruijsenaars in [28].

where

$$A^{(b)}_{P_s} \begin{bmatrix} P_2 & P_3 \\ P_1 & P_4 \end{bmatrix} = \frac{16\cosh(\pi b(P_s \pm P_1 \pm P_2 + \frac{ib}{2}))\cosh(\pi b(P_s \pm P_3 \pm P_4 + \frac{ib}{2}))}{\sinh(2\pi b P_s)\sinh(2\pi b(P_s + \frac{ib}{2}))^2 \sinh(2\pi b(P_s + ib))}. \tag{2.9}$$

It will turn out that $D$ and $\tilde{D}$ are related to $H$ and $\tilde{H}$ by a change of normalization. Our first claim is that the series $f$ in (1.15) satisfies the four shift equations

$$D^{(b^{\pm 1})}_{P_s} f^{(b)}_{P_s,P_t} = 2\cosh(2\pi b^{\pm 1} P_t) f^{(b)}_{P_s,P_t}, \tag{2.10}$$

$$\tilde{D}^{(b^{\pm 1})}_{P_t} f^{(b)}_{P_s,P_t} = 2\cosh(2\pi b^{\pm 1} P_s) f^{(b)}_{P_s,P_t}. \tag{2.11}$$

Let us explain why it is so. Notice from (1.16) that we have $e^{\pm ib\partial_{P_s}}\alpha_l(b^{-1}, P_s) = \alpha_l(b^{-1}, P_s)$. Then, substitution of (1.15) into (2.10) with the plus sign leads to

$$e^{2i\pi P_s P_t} \sum_{k=0}^{\infty}\sum_{l=0}^{\infty} \alpha_l(b^{-1}, P_s)\Big[\alpha_{k+1}(b, P_s - ib) + \alpha_{k-1}(b, P_s + ib)A^{(b)}_{P_s}$$

$$+ \alpha_k(b, P_s)V^{(b)}_{P_s} - \alpha_{k-1}(b, P_s) - \alpha_{k+1}(b, P_s)\Big]e^{-2\pi kbP_t}e^{-\frac{2\pi l P_t}{b}} = 0.$$

We now claim that the family of coefficients $\alpha_n$ defined in (1.16) is the unique solution of the shift-recurrence relation

$$\alpha_{k+1}(b, P_s - ib) + \alpha_{k-1}(b, P_s + ib)A^{(b)}_{P_s} + \alpha_k(b, P_s)V^{(b)}_{P_s} - \alpha_{k-1}(b, P_s) - \alpha_{k+1}(b, P_s) = 0 \tag{2.12}$$

which satisfies $\alpha_n(b, P_s) \to 0$ as $\mathrm{Re}(P_s) \to +\infty$. We conclude that the series f satisfies the two shift equations (2.10), since $f$ is invariant under $b \to b^{-1}$. Finally, we verified numerically in the domain of convergence of the series that

$$f^{(b)}_{P_s,P_t}\begin{bmatrix} P_2 & P_3 \\ P_1 & P_4 \end{bmatrix} = f^{(b)}_{P_t,P_s}\begin{bmatrix} P_2 & P_1 \\ P_3 & P_4 \end{bmatrix}. \tag{2.13}$$

Therefore, $f$ also verifies (2.11). Let us remark that we do not have an analytic proof that $\alpha_n$ in (1.16) satisfies (2.12), however, we verified it numerically up to high order.

## 2.2 Construction of $F$ and $\hat{F}$ from the series

The first important observation is that the two series

$$f^{(b)}_{P_s,P_t}\begin{bmatrix} P_2 & P_3 \\ P_1 & P_4 \end{bmatrix} \qquad \text{and} \qquad f^{(ib)}_{iP_s,iP_t}\begin{bmatrix} iP_2 & iP_3 \\ iP_1 & iP_4 \end{bmatrix}$$

satisfy the same shift equations, because the shift operators $D$ and $\tilde{D}$ are invariant under the transformations $b \to ib$, $P_i \to iP_i$, $i = 1,2,3,4,s,t$. The second key observation is that we have the following transformations between shift operators

$$K^{(b)}_{P_s,P_t}\begin{bmatrix} P_2 & P_3 \\ P_1 & P_4 \end{bmatrix} D^{(b)}_{P_s}\begin{bmatrix} P_2 & P_3 \\ P_1 & P_4 \end{bmatrix}\left(K^{(b)}_{P_s,P_t}\begin{bmatrix} P_2 & P_3 \\ P_1 & P_4 \end{bmatrix}\right)^{-1} = H^{(b)}_{P_s}\begin{bmatrix} P_2 & P_3 \\ P_1 & P_4 \end{bmatrix}, \tag{2.14}$$

$$K^{(b)}_{P_s,P_t}\begin{bmatrix} P_2 & P_3 \\ P_1 & P_4 \end{bmatrix} \tilde{D}^{(b)}_{P_t}\begin{bmatrix} P_2 & P_3 \\ P_1 & P_4 \end{bmatrix}\left(K^{(b)}_{P_s,P_t}\begin{bmatrix} P_2 & P_3 \\ P_1 & P_4 \end{bmatrix}\right)^{-1} = \tilde{H}^{(b)}_{P_t}\begin{bmatrix} P_2 & P_3 \\ P_1 & P_4 \end{bmatrix}, \tag{2.15}$$

as well as

$$\hat{K}^{(ib)}_{iP_s,iP_t} \begin{bmatrix} iP_2 & iP_3 \\ iP_1 & iP_4 \end{bmatrix} D^{(b)}_{P_s} \begin{bmatrix} P_2 & P_3 \\ P_1 & P_4 \end{bmatrix} \left( \hat{K}^{(ib)}_{iP_s,iP_t} \begin{bmatrix} iP_2 & iP_3 \\ iP_1 & iP_4 \end{bmatrix} \right)^{-1} = H^{(b)}_{P_s} \begin{bmatrix} P_2 & P_3 \\ P_1 & P_4 \end{bmatrix}, \qquad (2.16)$$

$$\hat{K}^{(ib)}_{iP_s,iP_t} \begin{bmatrix} iP_2 & iP_3 \\ iP_1 & iP_4 \end{bmatrix} \tilde{D}^{(b)}_{P_t} \begin{bmatrix} P_2 & P_3 \\ P_1 & P_4 \end{bmatrix} \left( \hat{K}^{(ib)}_{iP_s,iP_t} \begin{bmatrix} iP_2 & iP_3 \\ iP_1 & iP_4 \end{bmatrix} \right)^{-1} = \tilde{H}^{(b)}_{P_t} \begin{bmatrix} P_2 & P_3 \\ P_1 & P_4 \end{bmatrix}. \qquad (2.17)$$

These transformations are readily verified using the shift identities for the double Gamma function

$$\frac{\Gamma_b(z+b)}{\Gamma_b(z)} = \frac{\sqrt{2\pi}\, b^{bz-\frac{1}{2}}}{\Gamma(bz)}, \qquad (b \to b^{-1}). \qquad (2.18)$$

We then conclude that $\mathbf{F}^{+,(b)}_{P_s,P_t} \begin{bmatrix} P_2 & P_3 \\ P_1 & P_4 \end{bmatrix}$ and $\hat{\mathbf{F}}^{+,(\beta)}_{p_s,p_t} \begin{bmatrix} p_2 & p_3 \\ p_1 & p_4 \end{bmatrix}$ satisfy the same shift equations (2.4) and (2.5). Finally, we arrive at the claims (1.9) and (1.10) by symmetrizing with respect to $P_s$.

Let us finally mention that we added a phase in (1.13) so that our formula (1.9) reduces exactly to Ruijsenaars' asymptotic result [28] when only the first term in the series $f$ is kept. We believe this requirement eliminates the need to study the shift equations in the external momenta satisfied by the fusion kernel [4]. Moreover, we added a similar phase and a factor -i in (1.14) so that our proposals behave under Virasoro-Wick rotations just like in [23].

## 2.3   Comparing the coefficients $K$ and $\hat{K}$ by analytic continuation

The prefactor $K^{(b)}_{P_s,P_t} \begin{bmatrix} P_2 & P_3 \\ P_1 & P_4 \end{bmatrix}$ in (1.13) does not admit an analytic continuation to the regime $b^2 \in \mathbb{R}_-$, that is $c \leq 1$, since the function $\Gamma_b$ is not defined at these values. Similarly, the prefactor $\hat{K}^{(ib)}_{iP_s,iP_t} \begin{bmatrix} iP_2 & iP_3 \\ iP_1 & iP_4 \end{bmatrix}$ (1.14) does not admit an analytic continuation to $b^2 \in \mathbb{R}_+$. However, both $K$ and $\hat{K}$ are well-defined for complex values of $b^2$, so they can be compared in this regime (note that the same argument was used by Zamolodchikov in [32] to compare the structure constants of Liouville theory at $c \leq 1$ and $c \geq 25$). Let $b^2 \in \mathbb{C}$ such that $\mathrm{Im}\,(b^2) < 0$. Then, $K^{(b)}_{P_s,P_t} \begin{bmatrix} P_2 & P_3 \\ P_1 & P_4 \end{bmatrix}$ can be analytically continued to the point $\beta = ib$ without crossing the half-line $b^2 \in \mathbb{R}_-$. Moreover, utilizing the reflection formula [2, Equations (8-9)] for a function closely related to $\Gamma_b$ together with the definition $\Gamma_b(z) = \frac{\Gamma_2(z,b,b^{-1})}{\Gamma_2(\frac{Q}{2},b,b^{-1})}$, we have

$$\Gamma_b(z)\Gamma_{ib}(-iz+ib) = e^{-\frac{i\pi(Q-2z)^2}{16}} \frac{\left( -e^{i\pi b^{-2}}, e^{2i\pi b^{-2}} \right)_\infty}{(e^{2i\pi b^{-1}z}, e^{2i\pi b^{-2}})_\infty}. \qquad (2.19)$$

This identity can be used to express the ratio $K/\hat{K}$ in terms of q-Pochhammer symbols. More precisely, denoting $t(z) := \left( e^{2i\pi b^{-1}z}, e^{2i\pi b^{-2}} \right)_\infty^{-1}$, we find that

$$
\frac{K^{(b)}_{P_s, P_t} \begin{bmatrix} P_2 & P_3 \\ P_1 & P_4 \end{bmatrix}}{\hat{K}^{(ib)}_{iP_s, iP_t} \begin{bmatrix} iP_2 & iP_3 \\ iP_1 & iP_4 \end{bmatrix}} = e^{2i\pi \left( P_1^2 + P_2^2 + P_3^2 + P_4^2 + \frac{1+b^2+b^{-2}}{4} \right)}
\tag{2.20}
$$
$$
\times \frac{t(2iP_s)t(Q+2iP_s)}{t(-2iP_t)t(Q-2iP_t)} \frac{t(\frac{Q}{2}-iP_t \pm iP_2 \pm iP_3)t(\frac{Q}{2}-iP_t \pm iP_1 \pm iP_4)}{t(\frac{Q}{2}+iP_s \pm iP_2 \pm iP_1)t(\frac{Q}{2}+iP_s \pm iP_3 \pm iP_4)}.
$$

## 2.4 Domain of convergence of the series $f$

We have not analytically determined the convergence domain of the series $f$, and our numerical checks were conducted within a parameter range where the series $f$ is observed to converge.

We now exhibit a region of the parameter space where the series is likely to converge. We first restrict $b \in \mathbb{R}$ such that $b^2 \notin \mathbb{Q}$ and such that $b^2$ is not a Liouville number, that is,

$$
\exists m, q_0 \in \mathbb{N}, \qquad \forall (p,q) \in \mathbb{Z} \times \mathbb{Z}_{\geq q_0}, \qquad \left| b^2 - \frac{p}{q} \right| > \frac{1}{q^m}.
\tag{2.21}
$$

The assumption that $b^2 \notin \mathbb{Q}$ implies that $\sin(n\pi b^2) \neq 0$, whereas the assumption that $b^2$ is not a Liouville number implies that $\frac{1}{\sin(n\pi b^2)}$ admits a polynomial bound as $n \to \infty$ [20][6].

Next, we require that $P_i \in i\mathbb{R}$ for $i = 1, 2, 3, 4$ so that the coefficients $\phi_{nn}$ in (1.17) do not diverge exponentially as $n \to \infty$. Then, for $P_s \in \mathbb{R}$ we expect that the series $f^{(b)}_{P_s, P_t} \begin{bmatrix} P_2 & P_3 \\ P_1 & P_4 \end{bmatrix}$ converges in the domain $D = D_+ \cup D_-$ where

$$
D_\pm = \{ P_t \in \mathbb{C} \mid |\exp(-2\pi b^{\pm 1} P_t)| < 1 \}.
$$

We finally notice that in Liouville theory for $c < 1$ we have $\beta \in \mathbb{R}$ and the spectrum of the theory imposes that $p_t, p_s \in i\mathbb{R}$ [22]. Thus we have $|\exp(-2\pi \beta^{\pm 1} p_t)| = 1$, and the series diverges. However, in the fusion transformation (1.5) a nonzero real value $\Lambda$ is added to $p_t$. We can choose it to be positive so that the series converges.

## 2.5 $W(D_4)$ invariance of the series $f$

It was observed in [12] that the Virasoro conformal blocks possess discrete symmetries under three flips $s_i : P_i \to -P_i$ for $i = 1, 2, 3$ and under the so-called Regge-Okamoto transformation $s_\delta : P_i \to P_i - \delta$ with $\delta = \frac{1}{2} \sum_{i=1}^4 P_i$. These four transformations generate the Weyl group $W(D_4)$ of the Lie algebra of type $D_4$ (notice that the fourth flip $s_4$ is a certain product of the generators [12]).

---

[6]We thank Sylvain Ribault for suggesting the importance of $b^2$ not being a Liouville number.

It can readily be observed that the shift operators $D$ and $\tilde{D}$ in (2.7) and (2.8) – which admit the series $f$ as a joint eigenfunction – are $W(D_4)$-invariant (this was noticed first by Ruijsenaars in [28]). Although this does not necessarily imply that $f$ also has this invariance, we expect it to be true. In fact, the coefficient $\alpha_1$ in (1.19) is clearly $W(D_4)$-invariant, and we verified numerically that all $\alpha_n$ up to high order $n$ also have this invariance.

## 3    Consistency checks

### 3.1    The cases $c = 1$ and $c = 25$ with special external momenta

We now show that for special external momenta, namely $\Delta(P_i) = \frac{15}{16}$ (resp. $\Delta(P_i) = \frac{1}{16}$), the fusion kernels $\mathbf{F}$ for $c \to 25$ (resp. $\hat{\mathbf{F}}$ for $c \to 1$) have well defined limits which correspond to the known formulas [23]. More precisely, we have

$$\lim_{b \to 1} \mathbf{F}^{\pm,(b)}_{P_s,P_t} \begin{bmatrix} \frac{ib}{4} & \frac{ib}{4} \\ \frac{ib}{4} & \frac{i}{2b} - \frac{ib}{4} \end{bmatrix} = \mp i \frac{P_t}{P_s} 16^{P_s^2 - P_t^2} e^{\pm 2i\pi P_s P_t}, \tag{3.1}$$

$$\lim_{\beta \to 1} \hat{\mathbf{F}}^{\pm,(\beta)}_{p_s,p_t} \begin{bmatrix} \frac{i\beta}{4} & \frac{i\beta}{4} \\ \frac{i\beta}{4} & \frac{i}{2\beta} - \frac{i\beta}{4} \end{bmatrix} = 16^{p_t^2 - p_s^2} e^{\pm 2i\pi p_s p_t}, \tag{3.2}$$

which implies that

$$\lim_{b \to 1} \mathbf{F}^{(b)}_{P_s,P_t} \begin{bmatrix} \frac{ib}{4} & \frac{ib}{4} \\ \frac{ib}{4} & \frac{ib}{4} - \frac{i}{2b} \end{bmatrix} = \frac{P_t}{P_s} 16^{P_s^2 - P_t^2} \sin(2\pi P_s P_t), \tag{3.3}$$

$$\lim_{\beta \to 1} \mathbf{F}^{(\beta)}_{p_s,p_t} \begin{bmatrix} \frac{i\beta}{4} & \frac{i\beta}{4} \\ \frac{i\beta}{4} & \frac{i}{2\beta} - \frac{i\beta}{4} \end{bmatrix} = 16^{p_t^2 - p_s^2} \cos(2\pi p_s p_t). \tag{3.4}$$

The cases $c = 25$ and $c = 1$ are proved in the same way, hence let us focus on $c = 25$. Notice that it is a priori puzzling that we do not send all $P_i$'s to $ib/4$. The key is that by continuity, it should not matter how we approach the values $P_i = i/4$. However, we observed that the way (3.3) approaches the limit makes the computation trivial, as we see below.

We consider the limit (3.3) of $\mathbf{F}^{+,(b)}$, since the case of $\mathbf{F}^{-,(b)}$ consists of sending $P_s \to -P_s$. The crucial observation is that all coefficients $\phi_{nn}(P_1, P_2, P_3, P_4)$ and $\phi_{nn}(P_1, P_2 + \frac{i}{2b}, P_3 + \frac{i}{2b}, P_4)$, as well as their $b \to b^{-1}$ counterparts, vanish for all $n > 0$ at $P_1 = P_2 = P_3 = \frac{ib}{4}$ and $P_4 = \frac{ib}{4} - \frac{i}{2b}$. This implies that at these values $\alpha_{>0}(b, P_s) = \alpha_{>0}(b^{-1}, Ps) = 0$. Therefore, only the terms $k = l = 0$ in the sum (1.15) are nonzero and we obtain

$$f^{(b)}_{P_s,P_t} \begin{bmatrix} \frac{ib}{4} & \frac{ib}{4} \\ \frac{ib}{4} & \frac{ib}{4} - \frac{i}{2b} \end{bmatrix} = e^{2i\pi P_s P_t}. \tag{3.5}$$

It remains to compute the limit of the prefactor $K$. Utilizing the identity

$$\Gamma_1(z) = \frac{(2\pi)^{\frac{z-1}{2}}}{G(z)} \tag{3.6}$$

where $G(z)$ is the Barnes G-function, we obtain

$$\lim_{b\to 1} K^{(b)}_{P_s,P_t} \begin{bmatrix} \frac{ib}{4} & \frac{ib}{4} \\ \frac{ib}{4} & \frac{ib}{4} - \frac{i}{2b} \end{bmatrix}$$
$$= i(2\pi)^{-2i(P_s+P_t)} \frac{G(2-2iP_t)G(-2iP_t)}{G(2+2iP_s)G(2iP_s)} \frac{G(\frac{1}{2}+iP_s)^2 G(1+iP_s)^4 G(\frac{3}{2}+iP_s)^2}{G(\frac{1}{2}-iP_t)^2 G(1-iP_t)^4 G(\frac{3}{2}-iP_t)^2}.$$

Thanks to the doubling identity

$$G(2z) = C \, 2^{2z(z-1)} (2\pi)^{-z} G(\frac{1}{2}+z)^2 G(1+z)G(z) \tag{3.7}$$

where $C$ is an unimportant constant, we obtain

$$\lim_{b\to 1} K^{(b)}_{P_s,P_t} \begin{bmatrix} \frac{ib}{4} & \frac{ib}{4} \\ \frac{ib}{4} & \frac{ib}{4} - \frac{i}{2b} \end{bmatrix} = i 16^{P_s^2 - P_t^2} \frac{G(1+iP_s)^2}{G(2+iP_s)G(iP_s)} \frac{G(2-iP_t)G(-iP_t)}{G(1-iP_t)^2}. \tag{3.8}$$

It finally remains to utilize $G(1+z) = \Gamma(z)G(z)$, as well as $\Gamma(1+z) = z\Gamma(z)$ to obtain the desired result.

## 3.2 Crossing symmetry of Liouville theory

In this section we show that each component $\mathbf{F}^\pm$ and $\hat{\mathbf{F}}^\pm$ individually satisfies the crossing symmetry equations of Liouville theory on the four-point Riemann sphere. This is perhaps not surprising, since we verified numerically that each component individually satisfies the fusion transformation.

There exists a normalization of the primary fields in which the two and three-point correlation functions of Liouville theory for $c \in \mathbb{C}\backslash(-\infty, 1]$ are [19]

$$B^{b)}_P = \prod_\pm \Gamma_b(\pm 2iP)\Gamma_b(Q \pm 2P), \qquad C^{(b)}_{P_1,P_2,P_3} = \prod_{\pm,\pm,\pm} \Gamma_b(\tfrac{Q}{2} \pm P_1 \pm P_2 \pm P_3). \tag{3.9}$$

Similarly, for $c \leq 1$ we have

$$\hat{B}^{b)}_P = \frac{1}{4P^2 B^{(ib)}_{iP}}, \qquad \hat{C}^{(b)}_{P_1,P_2,P_3} = \frac{1}{C^{(ib)}_{iP_1,iP_2,iP_3}}. \tag{3.10}$$

The crossing symmetry equations for the four-point correlation function on the sphere for $c \in \mathbb{C}\backslash(-\infty, 1]$ and $c \leq 1$ can then be recast in terms of the fusion kernels:

$$\frac{C^{(b)}_{P_1,P_2,P_s} C^{(b)}_{P_s,P_3,P_4}}{B^{(b)}_{P_s}} \mathbf{F}^{(b)}_{P_s,P_t} \begin{bmatrix} P_2 & P_3 \\ P_1 & P_4 \end{bmatrix} = \frac{C^{(b)}_{P_2,P_3,P_t} C^{(b)}_{P_1,P_4,P_t}}{B^{(b)}_{P_t}} \mathbf{F}^{(b)}_{P_t,P_s} \begin{bmatrix} P_2 & P_1 \\ P_3 & P_4 \end{bmatrix}, \tag{3.11}$$

$$\frac{\hat{C}^{(b)}_{P_1,P_2,P_s} \hat{C}^{(b)}_{P_s,P_3,P_4}}{\hat{B}^{(b)}_{P_s}} \hat{\mathbf{F}}^{(b)}_{P_s,P_t} \begin{bmatrix} P_2 & P_3 \\ P_1 & P_4 \end{bmatrix} = \frac{\hat{C}^{(b)}_{P_2,P_3,P_t} \hat{C}^{(b)}_{P_1,P_4,P_t}}{\hat{B}^{(b)}_{P_t}} \hat{\mathbf{F}}^{(b)}_{P_t,P_s} \begin{bmatrix} P_2 & P_1 \\ P_3 & P_4 \end{bmatrix}. \tag{3.12}$$

Thanks to the duality (2.13), it can readily be verified that $\mathbf{F}^+$ and $\hat{\mathbf{F}}^+$ satisfy (3.11) and (3.12), respectively (note that for the latter we also need the shift identity (2.18)).

It then remains to show that $\mathbf{F}^-$ and $\hat{\mathbf{F}}^-$ satisfy the same equations. However, in this case the series do not trivially cancel. Keeping track of all factors, we find that $\mathbf{F}^-$ and $\hat{\mathbf{F}}^-$ respectively satisfy (3.11) and (3.12) if the series $f$ satisfies

$$\frac{f^{(b)}_{-P_s,P_t}\begin{bmatrix} P_2 & P_3 \\ P_1 & P_4 \end{bmatrix}}{f^{(b)}_{-P_t,P_s}\begin{bmatrix} P_2 & P_1 \\ P_3 & P_4 \end{bmatrix}} = u^{(b)}_{Ps}\begin{bmatrix} P_2 & P_3 \\ P_1 & P_4 \end{bmatrix} u^{(b)}_{-Pt}\begin{bmatrix} P_2 & P_1 \\ P_3 & P_4 \end{bmatrix}, \tag{3.13}$$

where

$$u^{(b)}_{Ps}\begin{bmatrix} P_2 & P_3 \\ P_1 & P_4 \end{bmatrix} = \frac{S_b(Q+2iP_s)}{S_b(Q-2iP_s)} \frac{S_b(\frac{Q}{2}-iP_1\pm iP_2-iP_s)S_b(\frac{Q}{2}-iP_3\pm iP_4-iP_s)}{S_b(\frac{Q}{2}-iP_1\pm iP_2+iP_s)S_b(\frac{Q}{2}-iP_3\pm iP_4+iP_s)}.$$

We verified this identity numerically up to high order in the domain of convergence of the series. Let us finally mention that an identity similar to (3.13) in the case of the one-point torus satisfied by the series $g$ (1.26) was proved by Ruijsenaars, see [27, Equation (2.29)].

## 3.3 The Virasoro-Wick rotation

Our proposals (1.9) and (1.10) for the Virasoro fusion kernels at $c > 25$ and $c < 1$ resemble each other. In fact, the components $\mathbf{F}^\pm$ and $\hat{\mathbf{F}}^\pm$ are related by a Virasoro-Wick rotation (2.6). More precisely, in this section we show that for $j = \pm$ we have

$$\mathcal{R}\mathbf{F}^{j,(b)}_{P_s,P_t}\begin{bmatrix} P_2 & P_3 \\ P_1 & P_4 \end{bmatrix} = -ji\hat{\mathbf{F}}^{j,(\beta)}_{p_s,p_t}\begin{bmatrix} p_2 & p_3 \\ p_1 & p_4 \end{bmatrix}. \tag{3.14}$$

The case $j = +$ is straightforward, because thanks to the identity (2.13) the two series cancel each other. It then suffices to handle the ratios of $\Gamma_{ib}$ functions by using the shift identity (2.18). The case $j = -$ necessitates an extra step. More precisely, we have

$$\frac{\mathcal{R}\mathbf{F}^{-,(b)}_{P_s,P_t}\begin{bmatrix} P_2 & P_3 \\ P_1 & P_4 \end{bmatrix}}{i\hat{\mathbf{F}}^{-,(\beta)}_{p_s,p_t}\begin{bmatrix} p_2 & p_3 \\ p_1 & p_4 \end{bmatrix}} = \frac{p_t}{p_s} \frac{K^{(\beta)}_{-p_t,p_s}\begin{bmatrix} p_2 & p_1 \\ p_3 & p_4 \end{bmatrix} f^{(\beta)}_{-p_t,p_s}\begin{bmatrix} p_2 & p_1 \\ p_3 & p_4 \end{bmatrix}}{\hat{K}^{(\beta)}_{-p_s,p_t}\begin{bmatrix} p_2 & p_3 \\ p_1 & p_4 \end{bmatrix} f^{(\beta)}_{-p_s,p_t}\begin{bmatrix} p_2 & p_3 \\ p_1 & p_4 \end{bmatrix}}. \tag{3.15}$$

We then use the identity (3.13) as well as the definition $S_\beta(z) = \Gamma_\beta(z)/\Gamma_\beta(\hat{Q}-z)$. The remaining part of the computation is straightforward: the resulting ratio of functions $\Gamma_\beta$ simplifies nicely thanks to (2.18).

## 3.4 The modular S matrix for the Virasoro characters

When the external field is the identity field, that is $P_0 = \frac{iQ}{2}$, the Virasoro modular kernel $\mathbf{M}$ becomes proportional to the modular S matrix for the Virasoro characters [3]. We now verify that the formula (1.20) satisfies this limit. In fact, when $P_0 = \frac{iQ}{2}$ (that is, $\alpha_0 = 0$) the coefficients $\mu$ in (1.28) satisfy $\mu_{\geq 1} = 0$. A straightforward computation then shows that

$$\mathbf{M}^{\pm,(b)}_{P_s,P_t}\begin{bmatrix} \frac{iQ}{2} \end{bmatrix} = \sqrt{2}\, e^{\pm 4i\pi P_s P_t}, \tag{3.16}$$

hence we have

$$\mathbf{M}_{P_s,P_t}^{(b)}\left[\frac{iQ}{2}\right] = \sqrt{2}\,\cos(4\pi P_s P_t),\qquad(3.17)$$

as expected from [3].

## Acknowledgements

We are grateful to thank Sylvain Ribault for illuminating discussions, and for his helpful comments on an earlier version of the draft and on the ancillary Jupyter notebook. We also thank Simon Ruijsenaars and Ioannis Tsiares for helpful discussions. We are supported by the Academy of Finland Centre of Excellence Programme grant number 346315 entitled "Finnish centre of excellence in Randomness and STructures (FiRST)".

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
