# Peer review of "On the Virasoro fusion and modular kernels at any irrational central charge"

_SciPost Physics, doi:SciPost Phys. 17, 138 (2024)_

## Round 3 · Referee Report · Sylvain Ribault (Referee 1) · 2024-8-29

Report

The Virasoro fusion and modular kernels are fundamental quantities in 2d conformal field theory. These quantities describe changes of bases of conformal blocks, and may be used for writing crossing symmetry and modular invariance equations -- the basic equations of the bootstrap approach.

These quantities are complicated functions of 7 and 4 parameters respectively, and obey many remarkable identities. It would be very useful to have explicit expressions of these quantities, although it is probably unrealistic to hope for a single expression that would lend itself to easy analytic manipulations, efficient numerical calculations, and have transparent symmetry properties. In the case of conformal blocks, we know various expressions of the same quantity, each one with its own advantages and limitations, including: a pedestrian sum over Virasoro descendants, two recursive formulas due to Zamolodchikov, and a sum over partitions from the AGT relation.

In the case of the fusion kernel, the situation greatly depends on the value of the central charge $c$: for $c\in \mathbb{C}-(-\infty, 1]$, we have 2 different integral formulas by Ponsot--Teschner and Teschner--Vartanov. And for $c\leq 1$, no explicit expression is known, except in special cases. However, many interesting CFTs have $c\leq 1$, starting with minimal models. In the case of the modular kernel, the situation is similar.

The article by Roussillon conjectures two series representations for the fusion kernel, valid for $c\in \mathbb{C}-(-\infty, 1]$ and $c\leq 1$. For $c\in \mathbb{C}-(-\infty, 1]$, this representation is numerically found to agree with the known integral representations. For $c\leq 1$, there is nothing to compare it with, so it is tested by directly checking the fusion relation for conformal blocks. The article also gives similar series representations for the modular kernel.

The series representations are obtained as solutions of certain shift equations that the kernels obey. These solutions are expected to be unique under certain analyticity assumptions. However, it is hard to prove convergence of the series representations, and therefore to control the analytic properties of the resulting kernels. This is why numerical tests are a crucial complement to solving the shift equations.

Additional evidence comes from the study of quite a few special cases where the kernels simplify, and from checks of remarkable identities that the kernels obey, but which are not manifest in the series representations. On balance, this article makes a very strong case for the conjectured series representations.

The article is generally well-organized and clearly written, although some local improvements are possible.

Remarks on substance:

  1. The relation between torus 1pt blocks and sphere 4pt blocks is attributed to Poghossian, but the formulas (1.7), (1.8) look much more like the relation by Fateev et al https://arxiv.org/abs/0902.1331, which differs from Poghossian's, and appeared earlier.

  2. In Section 1.4, it would be interesting to know more about the context and motivations of Ruijsenaars' work. In that work, how is $M$ defined, if not as a modular kernel? Why is this quantity interesting outside the context of CFT?

  3. The discussion of convergence in Section 2.3 is welcome, but the statements could be clarified. A number of assumptions on $b$ and $P_i$ are made: are these necessary for convergence, or just convenient for making a plausible argument? Based on analytic arguments and numerical observations, are there simple conjectures for regions in parameter space where the series converges or diverges? And would it not be simpler to study convergence of the modular kernel?

Clarity and other formal issues:

  1. In the introduction, when citing the uniqueness result of [4], both notations $b$ and $c$ are used, which is confusing.

  2. Typos in (1.19) and (3.13): $Ps \to P_s$.

  3. In the unnumbered table on page 6, what exactly is the "accuracy"? Rather than the real (or imaginary) part, why not compute the complex modulus of the difference?

  4. At the beginning of Section 1.3.2, the numerical value of the Ponsot--Teschner formula $0.308+5.22\times 10^{-22}$ is mysterious. Should we understand that the exact value is $0.308$, and the second term a numerical artefact?

  5. At the end of Section 1.3.2, the argument for the factor of 2 is obscure, a clarification would be welcome.

  6. After Eq. (2.9), it is a bit confusing to delay the relation between $D$ and $H$ until Section 2.2, rather than just write (2.14)-(2.17) there. Also, it could be stated more explicitly that (2.10), (2.11) are equivalent to (2.4), (2.5).

  7. At the end of Section 2.1, "to high order" could be made more quantitative.

  8. At the end of Section 2.2, for the Virasoro--Wick rotation, it would be more convenient to cite Eq. (2.6) rather than ref. [23].

  9. At the end of Section 2.3, it would be interesting to comment on the result (2.20), and explain why this result is useful or interesting.

  10. In Eq. (3.8), it might be good to write the simple end result for the limit of $K^{(b)}$, in addition to the ratio of $G$ functions that will eventually simplify.

NB: As a matter of principle, I have no recommendation about publication in this or that journal. Since it is necessary to select a recommendation for submitting a report, I picked the first one in the list.

Recommendation

Publish (surpasses expectations and criteria for this Journal; among top 10%)

  • validity: -
  • significance: -
  • originality: -
  • clarity: -
  • formatting: -
  • grammar: -

Author:  Julien Roussillon  on 2024-10-10  [id 4851]

(in reply to Report 1 by Sylvain Ribault on 2024-08-29)

Dear Prof. Ribault,

Thank you very much for your valuable report. Please find an answer to each of your recommendations below.

For the remarks on substance:

1) I added the reference of Fateev and al just after (1.7)-(1.8). 2) I added Section 1.4 which discusses the relation to Ruijsenaars' work. 3) I rewrote slightly the paragraph on convergence of the series, because the previous statements appeared slightly imprecise and incorrect. There is now one conjecture regarding the existence of a nonzero radius of convergence for the series. I provide a brief argument for why I expect the conjecture to be true. In Section 1.5, which now contains numberings, I added some text in number 1 to motivate the conjecture. Finally, in Section 2.1 I added a discussion on Ribault and Tsiares' conjecture and on Eberhardt's proof of uniqueness of the solutions of the shift equations.

Clarity and other formal issues:

1) I used only the notation of c. 2) I rectified the typos (I also noticed the same typo after (1.30) in the previous version). 3) Accuracy was misleading, and as you suggested, I modified the two tables in page 6 and just before section 1.4 by computing the complex modulus of the difference. 4) There is a typo, I meant $5.22 \times 10^{-22}i$. 5) There was a factor 1/2 missing in the definition of the Ponsot-Teschner formula in (1.3), because I defined the fusion transformation by an integral over R, and not over R_+. Hence I added a factor 1/2 in (1.3). 6) I reorganized section 2.1 as you suggested, and I merged section 2.2 into it. The objective of the new section 2.1 is to show that the proposals for F and \hat F satisfy the shift equations. 7) I now stated that I verified it up to order 50. 8) I now cited (2.6) instead of ref. [13]. 9) I decided to remove the former section 2.3, because I couldn't find a clear motivation for computing the ratio K/\hat K. 10) In Equation (3.8) I added two lines including the final result of the computation.

In addition to these changes, I moved the subsection on Virasoro-Wick rotations in Section 2. I also deleted the sentence in the abstract mentioning the limit b^2 rational of the formulas (instead, I added a Section 2.4). Finally, I reduced the number of bullet points in Section 1.5 down to 3.

Kind regards,

Julien Roussillon

---

## Round 3 · Referee Report · Anonymous (Referee 2) · 2024-9-2

Weaknesses

  1. Low significance of the obtained results.
  2. Lack of a proof of convergence of the obtained series representation of the crossing kernel.

Report

The integral kernel of the crossing transformation for the four-point correlation function studied in this work is an important object in two-dimensional conformal field theory. The work also contains a number of original results. However, I have several critical comments about it.

First and foremost, I would like to point out that the possibility of numerically, and therefore necessarily approximately, determining the values of the distribution represented by the studied object for specific argument values — which, in my opinion, is how the representation in the form of a series should be viewed — is not particularly significant. For proving the consistency of a conformal field theory model, what is important are the general analytical properties of this distribution, more precisely, the orthogonality relations it satisfies, which, by the way, the Author is aware of. These properties are known, and the reviewed work does not introduce anything new in this context.

A major drawback of the obtained result is the lack of proof of convergence for the series discussed in the work, as well as the lack of discussion on the range of arguments for which the series is convergent. One of the consequences of not providing such a proof is the inability to discuss the positions and types of singularities of the analyzed integral kernel.

The work should also be considered incomplete, as it contains — as the Author admits — statements that are merely hypotheses.

Recommendation

Reject

---

## Round 3 · Referee Report · Anonymous (Referee 3) · 2024-9-30

Report

Crossing kernels are fundamental kinematical objects in conformal field theory. They describe how conformal blocks are mapped to each other under change of OPE channel, and have played a central role in recent analytic approaches to the conformal bootstrap in two dimensions. A formula for the crossing kernel for Virasoro sphere four-point conformal blocks with central charge $c$ in the region $c\in\mathbb{C}\setminus(-\infty,1]$ is remarkably known in closed form due to work of Ponsot and Teschner, but no general formula is known in the case $c\leq 1$ and the Ponsot-Teschner formula cannot be analytically continued to this region.

The main result of this paper is to propose an infinite series representation, with recursively determined coefficients, for the Virasoro sphere four-point (and torus one-point) crossing kernels. The author argues for this series representation by demonstrating that it solves a shift equation that is a renormalization of that satisfied by the Ponsot-Teschner kernel (which itself descends from the pentagon equation). This also leads to an unfamiliar infinite series representation for the crossing kernel in the unproblematic region $c\in\mathbb{C}\backslash (-\infty, 1]$, whose equality with the Ponsot-Teschner kernel the author verifies numerically. The author verifies their proposal by showing that it numerically reproduces the known formulas at special values of the central charge and external conformal weights, and essentially, by numerically checking that it implements the crossing transformation on the conformal blocks. They also say that the formula numerically satisfies a functional relation that implies crossing symmetry of timelike Liouville CFT based on Zamolodchikov’s proposal for the $c\leq 1$ structure constants. None of these checked properties are manifest from the series representation.

To my mind, these checks are strong indications of the proposal’s correctness. However the infinite sum representation obscures the analytic structure of the crossing kernel, which will play an important role in most applications. An obvious target for future work will be to find a more efficient representation of the kernel akin to the Ponsot-Teschner formula (although an obstacle is that it must have weaker analyticity properties than the latter). The paper is well-written and the main result is novel and should find applications in the future. I recommend it for publication in SciPost.

Before publication I have some comments/questions that the author may wish to comment on: - In the discussion of the relation between the sphere four-point and torus one-point blocks, there should be a reference to the paper 0902.1331 by Fateev, Litvinov, Neveu and Onofri - The errors presented in the tables on pages 6 and 8 would be more meaningful if they were normalized in some way. Relatedly, the author should be more explicit about what is meant by "accuracy" in these tables. - I wonder if the combination $f^{(b)}{P_s,P_t}/ f^{(-ib)}$ (omitting the external operator labels) simplifies in the same way as $K$ prefactors. - I also wonder if the sphere four-point kernel simplifies in the case of pairwise identical external operators with exchange of the identity in the T-channel. In that case the Ponsot-Teschner kernel essentially reduces to the Liouville CFT structure constant.

Recommendation

Publish (meets expectations and criteria for this Journal)

  • validity: -
  • significance: -
  • originality: -
  • clarity: -
  • formatting: -
  • grammar: -

Author:  Julien Roussillon  on 2024-10-10  [id 4852]

(in reply to Report 3 on 2024-09-30)

Dear referee,

Thank you very much for your valuable report. I address below an answer to each of your recommendations.

  • I added this reference just after (1.7)-(1.8).
  • As you mentioned, "accuracy" was misleading. I rewrote the two tables by computing the absolute value of the difference of the two quantities.
  • Although it would certainly be interesting to be able to relate the two series, I could not find a clear motivation for computing the ratio K / \hat K. Since this is not an essential result of the paper, I decided to remove the former Section 2.3.
  • Thank you very much for this great suggestion. I added a Section 3.3 which describes this special case. I could recover exactly the expected result numerically. However, I do not have a complete proof, because it boils down to proving yet another mysterious identity satisfied by the series f.

Please find the response to Sylvain Ribault's report to see the other changes I made in the manuscript.

Kind regards,

Julien Roussillon

---

## Round 4 · Referee Report · Sylvain Ribault (Referee 1) · 2024-10-10

Report

The author has taken my remarks into account, and carefully answered my questions. The article's new version is significantly improved, both in form and in substance. I have no further improvements to suggest.

NB: As a matter of principle, I have no recommendation about publication in this or that journal. Since it is necessary to select a recommendation for submitting a report, I picked the first one in the list.

Recommendation

Publish (surpasses expectations and criteria for this Journal; among top 10%)

---

## Round 4 · Referee Report · Anonymous (Referee 3) · 2024-10-18

Report

The author has satisfactorily addressed my comments and questions on the previous version of their draft and I am now happy to recommend this paper for publication in SciPost.

Recommendation

Publish (meets expectations and criteria for this Journal)

---

## Round 4 · List of Changes

I modified the manuscript according to the recommendations of the referees. The list of changes can be found in the detailed answered to the referees' comments.

---

## Editorial Decision

published